# Physical activity for people living with cancer: Knowledge, attitudes, and practices of general practitioners in Australia

**Georgina Alderman[1,2,3], Richard Keegan[1,2,4], Stuart Semple[1,2,3,4], Kellie Toohey[1,2,3,4]***

**1** Discipline of Sport and Exercise Science, Faculty of Health, University of Canberra, Canberra, Australia, **2** Research Institute for Sport and Exercise, University of Canberra, Canberra, ACT, Australia, **3** Prehabilitation, Activity, Cancer, Exercise and Survivorship (PACES) Research Group, University of Canberra, Canberra, Australia, **4** Health Research Institute, University of Canberra, Canberra, Australia

* kellie.toohey@canberra.edu.au

**Data Availability Statement:** All relevant data are within the paper and its Supporting Information files.

## Abstract

### Background

Healthcare professionals' (Oncologists, doctors, and nurses) physical activity (PA) recommendations impact patients living with cancer PA levels. General practitioners (GPs) monitor the overall health of patients living with cancer throughout their treatment journey. This is the first study to explore GP's knowledge, attitudes and practices of PA for patients living with cancer.

### Methods

GPs who see patients living with cancer regularly (n = 111) completed a survey based on The Theory of Planned Behaviour (TPB). Participants (GP's) reported knowledge, attitudes, perceived behaviour control and subjective norms of PA within the cancer population. GP recommendation and referral rates of PA were reported. Principal component analysis was conducted to establish a set of survey items aligned to TPB constructs (attitude, subjective norms, perceived control), and multiple regression analyses characterised associations between these predictor variables and (a) recommendation; and (b) referral–of PA to cancer patients.

### Results

GPs (n = 111) recommended PA to 41–60% of their patients and referred 1–20% to PA programs. Multiple regression models significantly predicted the percent of patients recommended PA, p < .0005 adj. $R^2$ = 0.40 and referred PA, p < .0005, adj. $R^2$ = 0.21. GP attitudes and perceived behavioural control and GP's own activity levels were significant predictors of whether patients were recommended and referred for PA, p<0.05.

### Conclusion

GPs reported positive attitudes and perceptions towards promoting PA for their patients living with cancer. Despite having a positive correlation between PA recommendations and

**Funding:** The author(s) received no specific funding for this work.

**Competing interests:** The authors have declared that no competing interests exist.

referral rates, a gap was evident between GP's PA beliefs and their individual referral practices. More GP's willing to promote and refer their patients for PA, would improve the physical and mental health outcomes of the cancer population.

## Background

Globally, one in three people will to develop cancer during their lifetime [1]. Physical inactivity and poor dietary behaviours are significant contributing factors to cancer-related deaths [2]. Increased levels of physical activity (PA) has been shown to have a positive effect on reducing the development of secondary acute and chronic conditions [3, 4], including: comorbidity development [5]; cognitive impairment [6]); neurological issues [7]; cardiovascular and respiratory conditions [8]; lymphoedema [9]; cancer-related fatigue [10]; decreased bone mineral density [11]; and increased pain levels [12]. Although this evidence exists, PA levels within the cancer population are still sub-optimal, and reduced health outcomes are extremely common [13].

Healthcare professionals (HCPs) (oncologists, doctors and nurses) play a vital role in influencing patients' overall health and lifestyle behaviours [14]. It has been established that patients who have been told to be active by their HCPs' have improved exercise and adherence levels [15, 16]. HCPs position of authority creates a sense of respect from their patients and for the advice that they provide, meaning that this conversation about PA could improve the health of their patients.

Several psychosocial factors influence HCP behaviours–for example the Theory of Planned Behaviour (TPB) asserts that attitudes towards a behaviour, subjective appraisal of their group norms about the behaviour, and the perception of having sufficient control/choice to engage in the behaviour all precede intentions and actual behaviour. These variables have been shown to explain 31% of variance in observed behaviour in HCPs: a relatively strong finding compared to other theories [17]. The TPB is composed of three constructs (individuals' attitude, subjective norms, perceived behavioural control) [18]. The first construct is the individuals' attitude towards behaviour, in this case PA promotion having a positive or negative contribution to a cancer patients' care. The second construct is subjective norms which is defined by the perceived belief of what others think about a certain behaviour, in this case what other HCP's believe about the promotion of PA to their patients living with cancer. The final construct, titled perceived behavioural control, questions beliefs on the difficulty to perform the behaviour, in this case providing PA recommendations [17]. The three constructs are theoretically high predictors of individuals behaviour intentions and in turn lead to behaviour predictions [17].

The Clinical Oncology Society of Australia (COSA) released a position statement in 2018 encouraging all HCPs who are involved in the care of patients living with cancer to discuss and recommend "at least 150 minutes of moderate intensity aerobic exercise and two or three moderate intensity resistance exercise sessions each week" [19]. These recommendations match the guidelines for the general population [20]. In addition, the COSA statement suggests that HCPs should refer their patients for exercise advice to specialists such as accredited exercise physiologists, or physiotherapists with experience in cancer care [19]. PA recommendations from HCPs are appreciated and desired by cancer survivors [21–25] but typically this advice is not being given [16, 25, 26]. Jones, et al. (2002) looked at patient perspectives and reported that 97% of patients living with cancer would like PA to be discussed by their HCPs and 67%, believed that they should be referred for further specialised support, but there is still

no best practice evidence within more recent literature about who should have this discussion [27, 28]. PA is a modality of treatment, which cancer survivors can safely control themselves when prescribed [29]. Having the autonomy to control this during the unpredictable treatment period benefits prognosis and can improve patients' mental and physical health [30].

During treatment in Australia, it is reported that patients contact time is spent predominantly with oncology nurses and less time with their oncologists [31]. Health care professionals report that the discussion of PA is best provided by nurses [32–34]. Primarily in cancer care overall health and well-being is managed by GP's both during and post treatment [35]. However, the knowledge, attitudes and promotion of PA by GPs has not yet been studied. HCPs attitudes towards PA during the cancer treatment phase is evident [16, 24, 36], however, there is limited understanding of this during the post treatment phase [23, 37–39], GPs views and role within this area has not been explored. PA advice may be better-received post treatment when there is additional time to carry out this discussion with patients and they have had some recovery from the side effects of treatment [35]. GPs are well situated and could play a vital role in influencing cancer patient behaviours during this phase potentially positively impacting quality of life, and reducing co-morbidities and the recurrence of cancer [35].

A study conducted over a ten year period on Australian GPs, which tracked their attitudes and practices towards PA counselling in general population, demonstrated a significant increase in knowledge, confidence and the belief that it is part of their role to promote PA to their patients, however it was shown that GP's likelihood to promote physical activity was low [40]. Despite this, the TPB is based on the premise that attitudes, subjective norms, and perceived behavioural control affect a person's behavioural intention and this has a direct effect on behaviour [17]. GP's intention to engage in recommending PA to patients and referring patients living with cancer to exercise specialists is theoretically influenced by the value of the individuals position on the behaviour (attitudes), how it can be performed, the views of their fellow GP's (subjective norms) and the perception that the behaviour is within their control [17]. Therefore, the objective of this unique study was to establish predictors and determinants of GPs recommendations and referral behaviours for PA within the cancer population. Utilising the TPB, we hypothesise that; GP attitudes, perceived behavioural control and subjective norms will all be significant predictors of GPs recommendations and referral practices for PA in addition to their personal PA participation.

## Materials and methods

### Study instrument

The survey was developed based on the Theory of Planned Behaviour (TPB) Survey and items were located and adapted from previous research investigating PA attitudes and practices of other HCPs involved in cancer care [15, 16, 24, 32, 36, 41, 42]. The study instrument was generated on the Qualtrics questionnaire platform [43], and initial validation was conducted on an identical paper based version (S1 Appendix).

The first section comprised seven items relating to GPs' attitudes about PA during a cancer patient's treatment period and was depicted on the seven-point Likert scale. The second section contained four items and used a seven-point Likert scale to determine GPs' attitudes post treatment. The third section was composed of one ordinal item, four yes/no items and two Likert scale items, which assessed GPs' knowledge of PA for cancer specific and general populations. The fourth section of the survey was based on the promotion of PA. This section included one ordinal question ("what percentage of your patients living with cancer have you recommended PA to?"), and an ordinal question to rank the preference of PA modality. The fifth section of the survey was based on the referral practices of GPs. It comprised two ordinal

questions regarding percentage of clients which the GP refers to PA programs and treatment stage of preference to refer. A yes and no item was included to determine if GPs have PA resources readily available to them and a scalar question was used to determine the preferred HCP referral (options were physiotherapist, exercise physiologist, personal trainer, occupational therapist, sports medicine doctor and sport scientist).

GPs' personal PA levels were assessed with a yes/no question to understand if the GP participates in any regular structured exercise (>2 times per week) and a question to ask what best describes their activity level (seldom, moderate or vigorously active). This question was adapted from a single question study that assessed PA levels and was deemed both valid (weighted kappa = 0.75) and reliable (correlation coefficients r = 0.28 to 0.57) [44]. To enhance response rate, less mentally fatiguing questions were positioned at the end of the survey, i.e. demographics. This included age, gender, years of experience as a GP, and location of current practice (e.g., rural, sub-urban, urban). In addition, an exclusion criterion was introduced to ensure the GP had consulted a cancer patient in the last 12 months.

### Initial instrument validation

Prior to validation survey distribution, a total of nine individuals (five academics, two public officials and two accredited exercise physiologists) completed the survey. They were instructed to highlight items of confusion, construction problems, opinion on content and survey flow, and decipher any grammatical concerns. This comprised the content and construct validity portion of the validation process [45, 46]. Irrelevant questions were removed, additional questions of interest based upon expert opinion were added and sections were reordered to maximise response rate (demographic questions were moved to the end of the survey to minimise mental fatigue). Next a validation study was conducted with GPs located in Canberra ACT, Australia to determine the reliability and validity of the survey. This necessary step was identified as part of the systematic literature review conducted in preparing this project (Alderman et al., accepted for publication in October 2020). A minimum of 30 responses were required to satisfy criteria for first response validation [45]. In order to establish test-retest reliability a minimum of 10 retest responses were required. The analyses were conducted with SPSS V25 [47].

### Participants

Participants were recruited by private contact via phone and email of general practices. Only GPs were recruited due to the specificity of the target population for the overall research. Participants were recruited via yellow pages email dissemination across all states and territories of Australia (ACT, NT, TAS, WA, SA, NSW, VIC, QLD). In addition, the Australian College of Rural and Remote Medicine forwarded the survey link to their GP contacts.

### Study design

The study developed and validated, then distributed as cross-sectional national online survey of GPs who see patients living with cancer on a regular basis, to determine predictors of their recommendation and referral practices. An email was sent to GP clinics (to the practice managers) by the primary investigator stating the nature of the survey and for only GPs to partake in it and included an attached personalised video about the study to assist in increasing recruitment rate. GP's used the link in the email to access the survey, this link included the participant information, informed consent, and contact information for the primary investigator. If GP's chose to participate, they were required to tick a box which stated that they agreed to the informed consent prior to accessing the questions, this was an automated process. If they

chose to tick the no box, they were not eligible to access the survey. The study was approved by the University of Canberra Human Research Ethics Committee (HREC: 20191802).

### Data collection

The online Qualtrics platform was used to disseminate the survey and collect the data for this study. Independent anonymous links were sent via email which directed the participants to the survey. A follow up email was sent two, four and seven weeks after the initial email to remind clinics about the survey, survey cut off was established at eight weeks after initial email.

### Data analysis

The analyses were conducted on SPSS V25 [47] on completed surveys. Knowledge of PA was coded "correct" or "incorrect" to determine right PA guideline identification [20]. The conceptual framework of the instruments theoretical pathway is presented in Fig 1. The model was created based on The Theory of Planned Behaviour model [17]. Principal component analysis (PCA) was conducted to determine fit of the conceptual model allowing for item grouping of common themes for overall category score analysis. Bivariate Correlation was conducted to interpret correlation between overall variables and establish relationships. Multiple regression analysis was conducted to determine if components from the PCA and demographics were predictive characteristics of the percentage of patients recommended PA and referred by GPs. Pearson's product-moment correlation was run to assess the relationship between demographic variables, PCA conceptual components and dependant variables (percentage of patients recommended and referred PA).

## Results

### Initial instrument validation

A total of 31 GP responses were collected for the initial validation portion of this study (10 re-tests (of 31 invited to re-test) responses were collected response rate = 28.21%). All negative questions were reversed to establish the positive alternative for comparison purposes. Cronbach's alpha [45] was used to determine internal consistency ($\alpha > 0.7$) [14], questions assessing similar constructs should have a high level of internal consistency with Cronbach's alpha. The six questions on GP's attitudes to PA during treatment demonstrated a high level of internal consistency ($\alpha = 0.904$). 'Post treatment' construct consisted of three questions and had a high

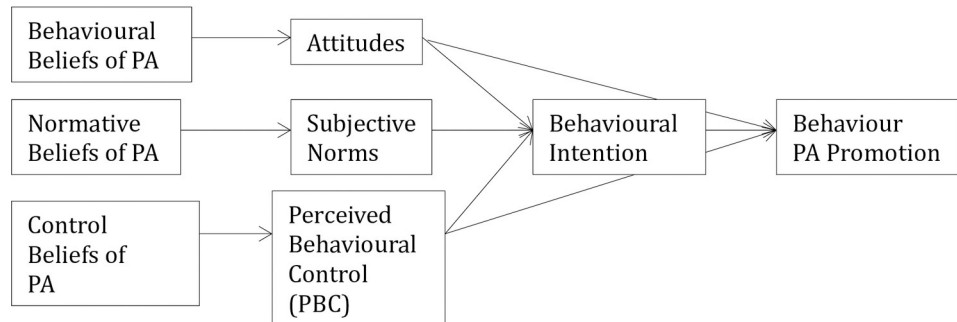

**Fig 1. Conceptual model.** PA = physical activity. Outlines the conceptual model the generated to construct the survey to align with the Theory of Planned Behaviour. Attitudes = attitudes in respect to PA promotion for the cancer population, Subjective norms = the views of their fellow GP, PBC = the perception that the behaviour is within their control (recommending/ referring PA to cancer patients).

internal consistency ($\alpha$ = 0.842). 'Evidence based practice' construct consisted of four questions which presented with a low internal consistency ($\alpha$ = 0.530). Upon removing Item 3, an acceptable internal consistence was reached of $\alpha$ = 0.701, however we determined that the removal of this question would impact the response of Item 4. Due to the categorisation of Item 3 and 4, these questions were split into the 'during' and 'post' treatment constructs, respectively and Cronbach's alpha was recalculated. 'During treatment' construct (7 items) $\alpha$ = 0.839. 'Post treatment' construct (4 items), $\alpha$ = 0.749. 'Evidence based practice' construct (2 items), $\alpha$ = 0.904. 'Promotion of physical activity' construct was not modified and consisted of four items ($\alpha$ = 0.805).

Test-retest results were collected two weeks post-initial response, with 10 participants responding to the second invitation. Each response was individually compared to determine inter-respondent variability and totalled. Variability averaged at 4.57%, hence presenting 95% overall consistency. A Pearson's product-moment correlation was run to assess the relationship between Likert responses in test-retest comparison [48, 49]. Spearman's correlation was additionally run [48, 49]. All items presented consistency ($p<0.05$), there were no significant outliers (S2 Appendix). An exact sign test was used to compare the differences in responses in 'Yes/ No' items within the survey [50]. Test-retest responses displayed no statistical difference between responses (0% variation evident- S2 Appendix). Overall, the survey demonstrated sufficient first and secondary response validation protocol [45, 46]. Previously stated amendments were conducted to satisfy validation criteria to prepare tool for national dissemination.

## Participant responses

A total of 128 general practitioners completed the survey; 97 online responses and 31 responses were included from the validation study. Only fully completed surveys were included, fifteen responses were excluded due to incomplete/ missing data, leaving a total of 113 completed for analysis. From total responses, 111 met inclusion criteria (having consulted a cancer patient in the last 12 months). The response rate, in relation to the targeted population, is unknown due to additional independent distribution of survey link from further organisations. This achieved sample 'response representation' displays a 9.2% margin of error [51]. This percentage is within the accepted margin of error of HCPs surveys (14%) [30], therefore offering an adequate representation of the views and practices of Australian GPs [52]. From 111 responses, 58% of GPs who completed the survey identified as females, examples of exercise intensity were provided [Table 1]. Average age of respondents was 49 (±10.4SD) years old with an average age of 19 (±10.4SD) years practicing.

## Principal component analysis

Principal component analysis (PCA) was run on survey items due to lack of validation measure of TPB components towards GP's recommendations of PA to patients living with cancer, hence examination of the conceptual model fitting was crucial [Table 2]. Review of the correlation matrix eliminated three questions that did not have at least one correlation coefficient greater than 0.3. The overall Kaiser-Meyer-Olkin (KMO) [53] was 0.831 with all individual KMO values greater than 0.6. Results of Barlett's test of sphericity were statistically significant ($p<0.0005$). PCA produced five components with values greater than one. Three components met interpretability criteria and were retained due to component loading. A three-component solution explained 59.8% of total variance. A Varimax orthogonal rotation was employed to aid interpretability. The rotated component matrix was consistent with component loadings [Table 2]. In relation to the Theory of Planned Behavior that was used to construct the scales: component one was consistent with GP attitudes, component two was consistent with

**Table 1. Participants' characteristics.**

| Characteristics | Number (%) |
| --- | --- |
| Sex | |
| Male | 47(42.34) |
| Female | 64(57.66) |
| Age (years) | |
| 26–35 | 13(11.71) |
| 36–45 | 28(25.23) |
| 46–55 | 38(34.23) |
| 56–65 | 25(22.52) |
| Over 65 | 7 (6.31) |
| Participate in regular PA | |
| Yes | 77 (69.37) |
| No | 64(30.63) |
| Description of activity level | |
| Vigorously active | 5(4.50) |
| Moderately active | 55(49.55) |
| Seldomly active | 51(45.95) |
| How many years practicing as a GP (years) | |
| 0–10 | 33(29.73) |
| 11–20 | 26(23.42) |
| 21–30 | 38(34.23) |
| 31–40 | 12(10.81) |
| 41+ | 2(1.80) |
| Location of practice | |
| Urban | 58(52.25) |
| Sub-urban | 33(29.73) |
| Rural | 20(18.02) |

perceived behavioral control, and component three with subjective norms. Where cross load-ing was identified, items were either assigned to the component category to which they showed the highest loading [17].

Preliminary analyses showed a linear relationship with percentage recommended PA with; percentage of patients referred PA, participation in structured PA, gender, attitudes, perceived behavioural control and norms as assessed by Shapiro-Wilk's test (p>0.05)–test of normality of data, and there were no outliers [Table 3]. Percentage of patients recommended PA pre-sented a linear relationship with; participation in structured PA, age, gender, attitudes, and perceived behavioural control (p>0.05), with no outlier's present.

## Multiple regression analysis

Multiple regression analyses were run to evaluate the relative contributions of attitudes, per-ceived behavioural control, subjective norms and other predictor variables on self-reported PA recommendation and referral behaviours of GPs, working with cancer patient's post-treat-ment. Gender, years practicing as a GP, practice location, personal activity levels, perceived behavioural control, attitudes of PA and subjective norms were assessed as predictor variables [Table 1]. Linearity assumptions were met via partial regression plots of studentized residuals against the predicted values. The data recommended PA and 'percentage of cases referred for PA' displayed an independence of residuals, assessed by a Durbin-Watson statistic of 1.907

**Table 2. Rotated structure matrix for PCA with varimax rotation of a three component questionnaire.**

| Items | Rotated Component Coefficients | | | |
|---|---|---|---|---|
| | C1-Attitude | C2- perceived behavioural control | C3-Norm | Communalities |
| Current evidence suggests regular physical activity is associated with reduced negative side effects of cancer treatment | **0.807** | | | 0.713 |
| Current evidence suggests that regular physical activity can improve quality of life of patients living with cancer | **0.729** | | | 0.671 |
| Most of my patients are capable of participating in physical activity during cancer treatment | **0.723** | | | 0.754 |
| Physical activity is beneficial during cancer treatment | **0.684** | | | 0.822 |
| Physical activity is important during cancer treatment | **0.657** | | | 0.849 |
| Patients would follow my advice, if I provided physical activity recommendations | **0.548** | | | 0.691 |
| My patients are amenable to receiving advice on the importance of increasing their physical activity levels | **0.533** | | | 0.770 |
| I feel confident in giving general advice to patients living with cancer about PA | | **0.838** | | 0.798 |
| Discussing physical activity with patients living with cancer is part of my role as a general practitioner | | **0.755** | | 0.749 |
| For me, providing a recommendation is easy | | **0.674** | | 0.581 |
| Physical activity is safe during cancer treatment | 0.502 | **0.518** | | 0.716 |
| Most patients believe they should be physically activity during cancer treatment | | | **0.867** | 0.777 |
| Fellow general practitioners think patients should participate in PA during cancer treatment | | | **0.770** | 0.741 |
| Other general practitioners believe it is part of their role to discuss physical activity with their patients | | | **0.687** | 0.757 |

Rotation Method: Varimax with Kaiser Normalization.

Rotation converged in 7 iterations.

**Note.** Major loadings for each item are in bold, only values < .5 are shown, cross loading was satisfied with largest number selection.

and 1.842 respectively. Homoscedasticity was observed under visual inspection of plotted studentized residuals versus unstandardized predicted values. No multicollinear results were

**Table 3. Bivariate correlation matrix.**

| Variable | 1 | 2 | 3 | 4 | 5 | 6 | 7 | 8 | 9 | 10 | 11 |
|---|---|---|---|---|---|---|---|---|---|---|---|
| 1. % Patients recommend PA to | - | | | | | | | | | | |
| 2. % Patients refer PA to | .550** | - | | | | | | | | | |
| 3. Participation in structured PA | .328** | .285** | - | | | | | | | | |
| 4. PA level | .063 | .063 | -.268** | - | | | | | | | |
| 5. Age | .087 | -.268** | .012 | -.111 | - | | | | | | |
| 6. Gender | .264** | .238* | .261** | -.048 | -.12 | - | | | | | |
| 7. Years of experience | .159 | .038 | .063 | -.037 | .896** | -.055 | - | | | | |
| 8. Location | -.027 | -.052 | .086 | .015 | .104 | .069 | .218* | - | | | |
| 9. Attitudes | .529** | .188* | .250** | -.025 | .247** | .192* | .256** | .027 | - | | |
| 10. PERCEIVED BEHAVIOURAL CONTROLPERCEIVED BEHAVIOURAL CONTROL | .594** | .374** | .260** | .01 | .220* | .17 | .236* | -.04 | .682** | - | |
| 11. Norms | .222* | .085 | .119 | -.027 | .14 | -.032 | .13 | -.123 | .424** | .404** | - |

PA- Physical activity, PERCEIVED BEHAVIOURAL CONTROL- Perceived behavioural control.

** Correlation is significant at the 0.01 level (2-tailed)

* Correlation is significant at the 0.05 level (2-tailed).

**Table 4. Summary of multiple regression analysis.**

| Variable | % Recommended PA | | | % Referred to PA | | |
|---|---|---|---|---|---|---|
| | B | SE | Beta | B | SE | Beta |
| Constant | -1.524 | 1.291 | | .668 | 1.274 | |
| Gender | .305 | .235 | .103 | .333 | .232 | .131 |
| Years practicing as a GP | .033 | .025 | .237 | .031 | .025 | .259 |
| Location | -.142 | .151 | -.074 | -.159 | .149 | -.097 |
| Description of PA level | .249 | .200 | .098 | .423 | .197 | .194 |
| **Participation in structured exercise** | .540 | .263 | **.170**[*] | .639 | .260 | **.234**[*] |
| **Attitudes** | .059 | .029 | **.221**[*] | -.032 | .028 | **-.137**[*] |
| **Perceived behavioural control** | .148 | .039 | **.394**[*] | .128 | .039 | **.396**[*] |
| Norms | -.023 | .036 | -.053 | -.015 | .035 | -.041 |

**Note.** B = unstandardized regression coefficient; SE = Standard error of the coefficient; Beta = Standardised coefficient

[*] = P<0.05.

present (tolerance values greater than 0.1). One studentized deleted residual was presented greater than ±3 (item number 42 = 3.1) but this did not impact additional results and so was retained. No leverage values greater than 0.2, and values for Cook's distance above 1 were evident. The multiple regression models significantly predicted the percent of patients living with cancer recommended PA F(9,101) = 9.129, P < .0005 adj. $R^2$ = 0.40 and referred to PA F (9,101) = 4.255, P < .0005, adj. $R^2$ = 0.21. Similarly, both items presented three variables that added statistical significance to the prediction, p<0.05. Regression coefficients and standard error can be seen in Table 4.

## Participant PA levels

General Practitioners' participation in structured PA was statistically significant to the prediction of percentage of patients living with cancer that were both recommended and referred PA (p<0.05). Descriptive statistics suggest 69.4% (n = 77) of GPs reported that they participated in structured PA, with the majority rating their PA levels at 'moderately active at least 30 minutes three times per week' (49.6%). In relation to knowledge of PA, 61.3% (n = 68) of GPs correctly identified the correct PA guidelines for general population. Only 3.6% noted that they had received PA training during their studies, 25.2% (n = 28) conducted additional training regarding PA knowledge of patients living with cancer, and 19.8% (n = 22) were aware of the Clinical Oncology Society of Australia (COSA) guidelines. In addition, only 31.5% (n = 35) of GPs had access to resources or were aware of PA services for patients living with cancer. Yet, 94.6% (n = 105) of GPs allocated exercise physiologists (59.5%) and physiotherapists (35.1%) as their HCP of choice to deliver PA guidance and services.

## Participant attitudes toward PA

General practitioners' attitudes towards PA for patients living with cancer were a significant predictor of the percentage of patients living with cancer recommended and referred for PA (p<0.05). Within this component, a high percentage of GPs had acknowledged that current evidence suggests that regular PA can both improve quality of life (92.8%) and reduce negative side effects of cancer specific treatment (89.2%). A large percentage of GPs recognised that PA is both important (91.0%) and beneficial (91.0%) to a cancer patient's journey. Seventy-two percent of GPs believed their patients to be capable of participating in PA and 79.3% stated that they believe their patients to be amenable to receiving PA advice. In addition, 64% of GPs

believe their patients would follow their advice if provided. Still, only 27% reported that their patients asked them about PA.

## Participant perceptions

The perceived ability to perform the behaviour was a significant predictor of both percentages of patients recommended and referred PA (p<0.05). Within this component, it was evident that 90.1% of GPs believe it to be safe for patients living with cancer to participate in PA. Sixty-seven percent of GPs believed providing PA recommendations was easy, and 77.5% were confident in giving recommendations (the highest modality of PA recommended being walking—49.5%). In addition to this, a high percentage of GPs believed that it was their role to discuss PA with their patients living with cancer (53.2%).

## Discussion

This is the first study of its kind to assess the knowledge, attitudes, and practices of Australian GPs regarding PA for the cancer population. Results offered some support for the TPB, but not all principles were supported. As hypothesised, both perceived behavioural control and attitudes were significant predictors of intentions to both recommend PA and refer patients to PA programs or for exercise specialist advice. Subjective norms, however, were shown to have no correlation with either behaviour, contradicting the hypotheses derived from the TPB. In addition, GPs personal PA participation predicted both the percentage of patients recommended to do PA and referred to PA programs or for further support by an exercise specialist.

The findings of this study suggested that the GPs were recommending PA to 41–60% of their patients living with cancer, indicating that 40–59% of patients were not receiving any discussion about PA. This number further decreases with GPs only referring 1–20% of patients living with cancer to an exercise program or to receive exercise specialist PA advice. These findings are consistent with similar studies looking at other HCPs (such as oncologists, oncology nurses and specialists) involved in cancer care [16, 32, 45, 54–57].

A large proportion of the surveyed GP population reported that they were, themselves, physically active (~70% and meeting the PA guidelines [Table 1]. This is substantially higher than reported in the general population in Australia (55.4%) [58]. In addition, similar studies that investigated oncology HCPs showed that >60% of practitioners were not meeting PA guidelines [16, 24, 27, 33, 36, 41, 59, 60]. This may suggest that GPs who responded to this survey report being more physically active than other HCPs report or perhaps those who responded were particularly interested in this topic because they were already regular exercises; those who didn't think it was important most likely didn't respond. This study builds upon the existing evidence showing that overall GP's personal PA levels played a significant role in the likelihood that they would recommend PA to their patients living with cancer [61] [Table 4]. Nonetheless, the current study is the only research conducted on GP's that has demonstrated a positive association between personal PA levels and PA referral practices [32, 36, 55, 59].

A positive relationship was also observed between PA knowledge and the promotion of PA [41], with >60% of GPs in the current sample correctly identifying the PA guidelines for the general population. This proportion is significantly higher in comparison to other HCPs involved in cancer care (<50%) [16, 57]. Accurate PA knowledge is arguably essential for informing best-practice. Accurate, in-depth PA recall to patients is positively associated with uptake and adherence levels of HCPs patients living with cancer [15, 16]. Despite 61% of GPs correctly identifying the general population PA guidelines, this study identified that GPs received minimal education on PA for the cancer population, both during university-training

and post-qualification professional education. The effect of PA education among GPs has not been thoroughly investigated. Identification that a minimal percentage of GPs were provided with PA education suggested that additional research should be conducted investigating the knowledge and education of GPs involved in cancer care in order to implement intervention-based studies at both an undergraduate and professional development level. This could assist in altering the training and applied practice of PA among GPs which would improve cancer patient outcomes.

In line with the TPB, the identified correlation between GPs attitudes and their PA recommendations and referral practices is consistent with existing research on oncologists in this area [41]. The current study shows that overall GPs attitudes of PA for patients living with cancer are significantly higher than other HCPs involved in cancer care. A high majority HCPs (oncologists, oncology nurses and specialists) involved in cancer care understand that PA improves the quality of life of their patients living with cancer [33, 34, 41]. Conversely, on average only 50% of HCPs recognise that regular PA is associated with a reduction in the negative side effects resulting from cancer treatment [34], compared to 89% of the GPs in the current study. In comparison to oncology HCPs, greater portions of GPs understand that PA is beneficial and important for patients living with cancer. General practitioners also recognise that patients living with cancer are capable of participating in PA, which had previously been noted as a sizeable barrier to PA recommendations in previous oncology HCP studies [24, 54, 55].

The perceived ability to control the implementation, perceived behavioural control of PA, as defined as a concept of TPB was a significant predictor of practice outcomes [Table 4]. These results are consistent with an international study investigating the knowledge, attitudes and practices of oncologists carried out in 2018 [41]. Further investigation on the individual items of the component showed a similar disconnection between the perceived behavioural control of GPs and other oncology HCPs. The current study identified that GPs have a significant understanding that PA is safe for patients living with cancer and that it is part of their role to provide these recommendations. Yet, a smaller percentage of GPs have the confidence to provide these recommendations to patients living with cancer and even fewer find it easy in comparison to percentage that thought it was their role [33, 62]. This percentage gap could be due to lack of specific GP education on PA recommendations for the cancer population.

In the current study, only 32% of GPs reported having access to resources about PA for patients living with cancer or specific PA programs. Studies conducted investigating patients living with cancer or PA programs indicated lack of referrals, with the average program occupancy rate approximately 70% across locations [63]. The difference between GPs access to resources and referrals and the researched occupancy rate for PA programs for cancer survivors implies a lack of awareness or additional barriers stopping them from referring. Further research should be conducted to see if the implementation of education tools within this population could increase cancer patients' participation in PA. A similar small but promising study was conducted on oncologists, they were provided with a 30-second education tool about PA recommendations. Results reported that patients PA increased by 3.4 MET hours per week in comparison to the control group [64].

As previously discussed, a larger scale study conducted over 1997-2007on Australian GPs, displayed a consistent incline in the knowledge, confidence, and perception that it was their role was to discuss PA. Despite this natural incline, no significant increase or correlation was seen in the promotion and practices of PA among participants [40]. Unlike this study, the current study demonstrated that the attitudes and perceived behavioural control of GPs were significant predictors of PA recommendations and referrals. The current study specifically looked at GPs knowledge, attitudes, and practices of PA specifically within the cancer population. Despite the predictive correlation between the TPB components and GPs practices, on

average greater than 80% of GPs have positive attitudes and perceived behavioural control towards PA in the cancer population. However, the average percentage of patients living with cancer recommended and referred to PA is substantially lower; 41–60% and 1–20% respectively. This inconsistency could reflect that the current study is potentially capturing some psychological predictors, however no other systemic, relational, resource and time-constraint predictors which could be of importance. Therefore, interventions targeting solely the GPs psychological predictors may not create a significant change. Thus, assessing the system holistically for future studies should be considered.

## Conclusion

It is well understood by GPs that it is part of their role to be promoting PA to their clients within the cancer population (>85%) and the general population [40]. With the consistent growth of research proving the benefits of PA for patients living with cancer with a reduction of adverse effects from cancer treatments, an emphasis needs to be placed on GPs to promote this message to their patients. Evidence suggests that one of the greatest limitations to increase the evidence in this space is the current lack of referrals from oncology HCPs and GP's into PA programs run by exercise specialists [29]. The current study suggests that enhancing the psychological aspects of GPs attitudes and perceived behavioural control towards PA could create a positive impact on the percentage of patients living with cancer both recommended and referred for PA. In addition to this, GPs are in a highly influential position to enhance PA referral rates due to their positive attitudes and knowledge of the important role of PA for the cancer population. GP's in Australia have the power to activate government funded (Medicare) 'rebate-able' programs with exercise specialists for patients living with cancer, meaning more patients could potentially have access to programs enhancing physical and mental health. Future studies and patient care policies should look at ways to integrate this information to enhance GP's involvement in the care of patients living with cancer throughout the cancer treatment and survivorship continuum, through specific cancer care education and training.

## Limitations

Despite the study findings, limitations of this study must be taken into consideration. The authors acknowledge that due to the study design consisting of a self-reported survey, a bias likely evident as a result of potential over reporting of personal PA levels by the GP's. In addition, response rate was limited and may not fully represent the overall Australian GP population. Assessing demographic data of the sample displayed a significant proportion of GPs reporting themselves to be meeting the PA guidelines. A bias is also likely due to the percentage of GPs self-reporting that they are significantly more active than the Australian population. Perhaps those who responded to the survey were particularly interested in this topic because they were already regular exercises; those who didn't think it was important may not have responded. Strengths of the study include, its integration of the conceptual model generated around the theory of planned behaviour, principal component analysis to test the fitting of the conceptual model, the survey validation and the systematic literature review conducted to highlight the gap in the literature. This was also the first study of its kind to look at GP's knowledge and practices of PA within the cancer population.

## Supporting information

**S1 Data.**
(XLSX)

**S1 Appendix. Survey**
(DOCX)

**S2 Appendix. Test-retest analysis procedure.**
(DOCX)

**S3 Appendix. PCA total variance SPSS output.**
(DOCX)

## Author Contributions

**Conceptualization:** Stuart Semple, Kellie Toohey.

**Data curation:** Richard Keegan.

**Formal analysis:** Georgina Alderman, Richard Keegan.

**Investigation:** Georgina Alderman, Kellie Toohey.

**Methodology:** Georgina Alderman, Richard Keegan, Stuart Semple, Kellie Toohey.

**Project administration:** Georgina Alderman.

**Resources:** Kellie Toohey.

**Supervision:** Richard Keegan, Stuart Semple, Kellie Toohey.

**Writing – original draft:** Georgina Alderman.

**Writing – review & editing:** Richard Keegan, Stuart Semple, Kellie Toohey.

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
