## [Decision Letter · Decision Letter 0]

17 Jul 2020

PONE-D-20-10478

Physical Activity for Cancer Patients: Knowledge, Attitudes, and Practices of General Practitioners in Australia

PLOS ONE

Dear Dr. Toohey,

Thank you for submitting your manuscript to PLOS ONE. After careful consideration, we feel that it has merit but does not fully meet PLOS ONE’s publication criteria as it currently stands. Therefore, we invite you to submit a revised version of the manuscript that addresses the points raised during the review process.

This is an important and timely paper.  The reviewers have made a number of suggestions, particularly around including more information in the analysis section, as well as explaining the TPB more clearly. 

We look forward to receiving your revised manuscript.

Kind regards,

Adam Todd, PhD

Academic Editor

PLOS ONE

Journal Requirements:

3. In your Methods section, please provide additional information about the participant recruitment method and the demographic details of your participants. Please ensure you have provided sufficient details to replicate the analyses such as: a) the recruitment date range (month and year) and b) a description of any inclusion/exclusion criteria that were applied to participant recruitment.

4. To comply with PLOS ONE submission guidelines, in your Methods section, please provide additional information regarding your statistical analyses. For more information on PLOS ONE's expectations for statistical reporting, please see https://journals.plos.org/plosone/s/submission-guidelines.#loc-statistical-reporting.

5. We ask that you please remove citations for unavailable and unpublished work, including manuscripts that have been submitted but not yet accepted (e.g., “in submission,” “data not shown”). Instead, include those data as supplementary material or deposit the data in a publicly available database.

7. Your ethics statement must appear in the Methods section of your manuscript. If your ethics statement is written in any section besides the Methods, please move it to the Methods section and delete it from any other section. Please also ensure that your ethics statement is included in your manuscript, as the ethics section of your online submission will not be published alongside your manuscript.

Reviewers' comments:

Reviewer's Responses to Questions

**Comments to the Author**

1. Is the manuscript technically sound, and do the data support the conclusions?

Reviewer #1: Yes

Reviewer #2: Yes

Reviewer #3: Yes

2. Has the statistical analysis been performed appropriately and rigorously? 

Reviewer #1: I Don't Know

Reviewer #2: I Don't Know

Reviewer #3: Yes

3. Have the authors made all data underlying the findings in their manuscript fully available?

Reviewer #1: Yes

Reviewer #2: Yes

Reviewer #3: Yes

4. Is the manuscript presented in an intelligible fashion and written in standard English?

Reviewer #1: No

Reviewer #2: Yes

Reviewer #3: Yes

5. Review Comments to the Author

Reviewer #1: PLOS ONE Review:

Physical Activity for Cancer Patients: Knowledge, Attitudes, and Practices of General Practitioners in Australia.

Abstract:

Line 69-70: was this in relation to the behaviour of promoting exercise or their actual exercise behaviour?

Line 71-72: be more specific. Fit of TPB to what? Associations of what?

Background:

Line 84: avoid ambiguous terms “are likely”…change to “will” if this is what the research is showing. Also, can you contextualize this geographically. Do you mean globally?

Line 89-90: reference this statement.

Line 94: add ‘exercise’ before ‘levels’

Line 99-100: not sure what you mean by “playing a strong predictive power for HCP’s compared to…” please clarify.

Line 100-101: reference this sentence “TPB is composed of…” (Ajzen article)

Line 101: change ‘constraint’ to ‘construct’

Line 103-104: wording needs updating grammatically

Line 112: Specify COSA acronym in line 109

Line 119: remove ‘of’

Line 115: Jones article is quite old, will likely be able to supplement with something more recent. Much has changed since 2002 in regard to oncology rehab, that is why I suggest this.

Line 124: change ‘reported’ to ‘report’

Line 124-125: again, can you contextualize this; where does that happen? This is not always the case in North America.

Line 126: change ‘to’ to ‘towards’

Line 129: missing word after ‘treatment (32)’

Line 133: missing a word after ‘decade’

Line 136-137: I would not say that the TPB is based on the premise that HCPs…The TPB is based on the premise that attitudes, subjective norms, and PBC affect a person’s behavioural intention and this has a direct effect on behaviour.

Line 138: Change to GP as you are using elsewhere in intro.

Overall, I think the TPB needs to be explained more clearly; specifically, the ‘intention’ component was largely missing and is very important to this theory.

Good ideas highlighted in the intro however.

Materials and Methods:

Missing information between ‘participant recruitment’ and ‘data collection’: Line 152-163

- More details are needed on the process after the recruitment email was sent; did those interested respond via email to XX.

- Who then emailed the survey to the GP’s??

- Why does the data collection section say it was also practice managers and clinics that were emailed? More information needed. Also in this section should reference ethics received.

Line 162: Can you give a few survey question examples within the manuscript? It would be interesting to see how your survey addressed each of the TPB components; perhaps a table with one column: TPB construct, another column “survey question(s) related to this construct”. I see you have It as an appendices but would be good to summarize within the written document.

Line 162: give more information in regard to the consent process. Did they have to sign this form prior to completing the survey, etc.

Line 162-163: confirm why follow up email was sent…if they did not complete the survey within two weeks. How many follow up emails were sent? Over what time frame? How long did respondents have to complete the survey?

Figure 1:

- Note: this conceptual model does not align. The overall behaviour is PA promotion but you are describing the attitude and subjective norm towards PA in general. These are two different behaviours. In the text you describe subjective norms related to PA promotion, but not in the figure. The attitude should be “attitudes in respect to PA promotion for the cancer population”…do they think it is important / helpful, etc. Must all link together and make sense.

Good description of initial validation.

Line 212-215 should go in ‘recruitment’ section. It is confusing how it is laid out…I think you are describing two separate studies. The methodology section should only be in reference to this study.

Paragraph Lines 219-231: you are presenting results here. Should go in results section. Organize methods / results more clearly to delineate the steps that were taken. It is all important, just needs to be organized more appropriately. Same thing for paragraph starting Line 233.

Results:

Line 248: how much needed to be complete to be included? All questions? Confirm in methods.

Line 255: add ‘exercise’ before intensity. Also, I think you are missing a statement about % who take part in exercise prior to this sentence.

Line 280-281; this should go in the methods. Use sub-headings in the results section to make clear to readers.

Line 293-295; can you describe this further…for example, what Gender? How many years practicing? This information will be interesting for readers but isn’t stated anywhere.

Line 311-312; what there a cut off value for this? How much did practitioners have to exercise to be more likely to predict…any?

Discussion:

Line 362-363: you should link this finding to limitations also; perhaps those who responded to your recruitment email were particularly interested in this topic because they were already regular exercises; those who didn’t think it was important didn’t respond. Add into limitations also.

Paragraph starting line 369. I think it was also interesting that only 19% were aware of the new COSA guidelines. This was a major publication in exercise oncology globally! So, the fact that GP’s within Australia are not aware of it is surprising.

Line 418 and throughout discussion: do not need to reference tables in discussion sections

Throughout discussion section you may want to refer to other studies looking at barriers/facilitators to exercise promotion for this population. What gets in the way and how does this relate to the TPB components? How can this be addressed?

Summary:

Overall, an interesting topic. Thank you for taking the time to conduct research in this area. Needs some edits for clarity before publication.

I am concerned around how the TPB was described and how the conceptual model include two different behaviours (PA and PA promotion)...this needs to be clarified.

More detailed and clear information needed in methods; results were presented in methods as well.

Increase clarify of results (give more specific information).

Reviewer #2: Thanks for the opportunity to review this paper, it was interesting to read. Please see comments which are a mix of relevant points and pedantry.

Abstract

I wonder if the term 'cancer patients' is appropriate these are people with a diagnosis of cancer and that label really does jar when I read it. We probably don't label patients as 'diabetics' or schizophrenics' any linger so maybe the term 'patients with cancer' or perhaps 'patients living with cancer', although not exactly mellifluous. would be more appropriate? Same comment applies to 'the cancer population' and other such terms

Background

Line 85 33% of cancer related deaths are attributable to physical inactivity....' I wonder is this statement too strong. I agree that physical inactivity and poor diet are contributory factors but as an absolute causal link it's very difficult to be that categorical. Could you say that they are significant contributory factors?

Line 93 -'It has been established that patients who have been told to be active by their HCPs’ have improved levels and adherence levels' this doesn't make sense to me improved levels and adherence level ? Im presuming you mean persistence with the exercise programme? Maybe reword ?

Line 103 - 'in this case what other HCP’s physical activity believe and promotion of PA to their cancer patients...' consider reword doesn't make sense should that beliefs are?

Line 105 repetition of 'perceived' not sure second one is necessary and removing would improve sentence flow

Line 106 should read 'individuals''?

Line 109 - define acronym if you are going to use it in the rest of the document Clinical Oncology Society of Australia (COSA)....

Line 112 should read 'the general population'

Line 118 '...treatment, which cancer...'

Line 122 - could these two sentence be combined, second sentence is partially repeating the first.

Line 126 is a possessive so should read 'HCPs'"

Line 133 should read '...Australian GPs. which ...'

Line 138 you have used GPs elsewhere, why write out in full here?

Methods

I'm not convinced that you need to specify whether data is ordinal, nominal or scalar. Its repetitive and doesn't help the flow of the text. I would argue anyone reading this is capable of that distinction and the nature of the data in statistical terms is immaterial, it is what it is.

Results

Line 258 - surprising that you only had people identify as male or female and there were no other options. Did you provide options for gender or was your question about sex ie male/female as options? If so this should read sex.

Line 296 should read recommended 'PA' ....

Line 311 - personal preference but as a reader where percentages are quoted I often like to see the number, just for context as I read so 69% (n=22) or 22 GPs (x%), whichever works best for you.

Discussion I would argue that Table 5 should be in the results section and discussed in the context of other literature in the discussion. If you feel uncomfortable with the results of the literature review from other HCPs then remove this and discussion in the discussion section. Feels unusual to see a table presented like this in the discussion.

Line 396 should read 'Conversely, on average....'

Line 390 should read ...in PA, which....'

Line 404 should read 'In the current study, only 32%....'

Line 430 is adverse effects a better term?

We would normally expect to see a conclusion in a study of this type, I think it would add clarity to the final part of this paper, which does wander a little bit.

Reviewer #3: I will focus on methods and reporting. Statistical analyses are appropriate.

Major

1) State clearly all information in the data analysis section, all used variables. How was correct and incorrect PA knowledge determined? Clarify what the outcome is and how it is recorded.

Minor

1) Abstract: more clarity on regression modelling, no information on covariates of interest

2) the survey is potential problematic, as the authors acknowledge. the generalisability of the survey is questionable.

3) in table 4 better to report CIs rather than SEs

4) Consider other graphical outputs to present your results and make them more accessible

6. PLOS authors have the option to publish the peer review history of their article (what does this mean?). If published, this will include your full peer review and any attached files.

Reviewer #1: No

Reviewer #2: No

Reviewer #3: No

---

## [Author Response · Author response to Decision Letter 0]

20 Sep 2020

20th September, 2020

Dear Reviewers,

Thank you very much for kindly taking the time to provide very helpful feedback on the manuscript. All suggested changes from the reviewers have been carefully considered and added to the manuscript. Below is the list of suggested changes along with the responses added to the manuscript (I have also included where these changes can be found in the manuscript).

Yours sincerely,

Dr Kellie Toohey (corresponding author)

Response to reviewers

Reviewers' comments:

Reviewer's Responses to Questions

Comments to the Author

1. Is the manuscript technically sound, and do the data support the conclusions?

Reviewer #1: Yes

Reviewer #2: Yes

Reviewer #3: Yes

2. Has the statistical analysis been performed appropriately and rigorously? 

Reviewer #1: I Don't Know

Reviewer #2: I Don't Know

Reviewer #3: Yes

3. Have the authors made all data underlying the findings in their manuscript fully available?

Reviewer #1: Yes

Reviewer #2: Yes

Reviewer #3: Yes

4. Is the manuscript presented in an intelligible fashion and written in standard English?

Reviewer #1: No

Reviewer #2: Yes

Reviewer #3: Yes

5. Review Comments to the Author

Reviewer #1: PLOS ONE Review:

Physical Activity for Cancer Patients: Knowledge, Attitudes, and Practices of General Practitioners in Australia.

Abstract:

Line 69-70: was this in relation to the behaviour of promoting exercise or their actual exercise behaviour?

• The behaviour of promoting exercise – we have made the following change to make it clearer: Participants (GP’s) reported knowledge, attitudes, perceived behaviour control and subjective norms of PA within the cancer population.

Line 71-72: be more specific. Fit of TPB to what? Associations of what?

• Thank you this has been corrected and now reads:

Principal component analysis was conducted to establish a set of survey items aligned to TPB constructs (attitude, subjective norms, perceived control), and multiple regression analyses characterised associations between these predictor variables and (a) recommendation; and (b) referral – of PA to cancer patients. 

Background:

Line 84: avoid ambiguous terms “are likely”…change to “will” if this is what the research is showing. Also, can you contextualize this geographically. Do you mean globally?

• Changed to: Globally, one in three people will develop cancer during their lifetime

Line 89-90: reference this statement.

• Added: {McTiernan, 2019 #3501} Physical Activity in Cancer Prevention and Survival: A Systematic Review

Line 94: add ‘exercise’ before ‘levels’

• Added: y their HCPs’ have improved exercise and adherence levels

Line 99-100: not sure what you mean by “playing a strong predictive power for HCP’s compared to…” please clarify.

• This has been re-worded to: Several psychosocial factors influence HCP behaviours – for example the Theory of Planned Behaviour (TPB) asserts that attitudes towards a behaviour, subjective appraisal of their group norms about the behaviour, and the perception of having sufficient control/choice to engage in the behaviour all precede intentions and actual behaviour. These variables have been shown to explain 31% of variance in observed behaviour in HCPs: a relatively strong finding compared to other theories (17)

Line 100-101: reference this sentence “TPB is composed of…” (Ajzen article)

• Added 

Line 101: change ‘constraint’ to ‘construct’

• Changed

Line 103-104: wording needs updating grammatically

• This has been corrected and now reads: subjective norms which is defined by the perceived belief of what others think about a certain behaviour, in this case what other HCP’s believe about the promotion of PA to their cancer patients.

Line 112: Specify COSA acronym in line 109

• This has been added to line 109 and now reads: The Clinical Oncology Society of Australia (COSA) released

Line 119: remove ‘of’

• Removed

Line 115: Jones article is quite old, will likely be able to supplement with something more recent. Much has changed since 2002 in regard to oncology rehab, that is why I suggest this.

• An updated references had been added and a bit more information has been added to the sentence to update the context: Jones, et al. (2002) looked at patient perspectives and reported that 97% of cancer patients would like PA to be discussed by their HCPs and 67%, believed that they should be referred for further specialised support, but there is still no best practice evidence within more recent literature about who should have this discussion

Line 124: change ‘reported’ to ‘report’

• Changed 

Line 124-125: again, can you contextualize this; where does that happen? This is not always the case in North America.

• This has been changed to give more context: During treatment in Australia, it is reported that patients contact time is spent predominantly with oncology nurses and less time with their oncologists

Line 126: change ‘to’ to ‘towards’

• Changed 

Line 129: missing word after ‘treatment (32)’

• This has been changed: PA advice may be better-received post treatment when there is additional time to carry out this discussion with patients and they have had some recovery from the side effects of treatment

Line 133: missing a word after ‘decade’

• Has been changed to: A study conducted over a ten-year period on Australian GPs

Line 136-137: I would not say that the TPB is based on the premise that HCPs…The TPB is based on the premise that attitudes, subjective norms, and PBC affect a person’s behavioural intention and this has a direct effect on behaviour.

• This has been changed – thank you for the wording it sounds a lot better.

Line 138: Change to GP as you are using elsewhere in intro.

• This has been changed

Overall, I think the TPB needs to be explained more clearly; specifically, the ‘intention’ component was largely missing and is very important to this theory.

• More detail has been added to explain the TBC more clearly.

Good ideas highlighted in the intro however.

Materials and Methods:

• The authors have re-arranged the materials and methods to improve flow of the manuscript.

Missing information between ‘participant recruitment’ and ‘data collection’: Line 152-163

• We have added study design and re-arranged the materials and methods section line 191

- More details are needed on the process after the recruitment email was sent; did those interested respond via email to XX.

• This has now been added with further details.

- Who then emailed the survey to the GP’s??

• This has been added in to line 194: . An email was sent to GP practices by the primary investigator stating the nature of the survey and for only GPs to partake in it and included an attached personalised video about the study to assist in increasing recruitment rate.

- Why does the data collection section say it was also practice managers and clinics that were emailed? More information needed. 

• This was how we gained access to the GP’s – the practice Managers then sent this on to the GP’s.

Also in this section should reference ethics received.

• This has been moved to study design section

Line 162: Can you give a few survey question examples within the manuscript? It would be interesting to see how your survey addressed each of the TPB components; perhaps a table with one column: TPB construct, another column “survey question(s) related to this construct”. I see you have It as an appendices but would be good to summarize within the written document.

• Thank you for the comment – table two has the rotation of a three-component questionnaire in regard to the TBC and the items (questions) and the weight of the TBC that they relate to.

Line 162: give more information regarding the consent process. Did they have to sign this form prior to completing the survey, etc.

• More details about this has been added to the study design line 190

Study Design

The study developed and validated, then distributed as cross-sectional national online survey of GPs who see patients living with cancer on a regular basis, to determine predictors of their recommendation and referral practices. An email was sent to GP clinics (to the practice managers) by the primary investigator stating the nature of the survey and for only GPs to partake in it and included an attached personalised video about the study to assist in increasing recruitment rate. GP’s used the link of the email to access the survey, this link included the participant information, informed consent, and contact information for the primary investigator. If GP’s chose to participate, they were required to tick a box which stated that they agreed to the informed consent prior to accessing the questions, this was an automated process. If they chose to tick the no box, they were not eligible to access the survey. The study was approved by the University of Canberra Human Research Ethics Committee (HREC: 20191802).

Line 162-163: confirm why follow up email was sent…if they did not complete the survey within two weeks. How many follow up emails were sent? Over what time frame? How long did respondents have to complete the survey?

• Thank you, more detail has been added to data collection

Data collection

The online Qualtrics platform was used to disseminate the survey and collect the data for this study. Independent anonymous links were sent via email which directed the participants to the survey. A follow up email was sent two, four and seven weeks after the initial email to remind clinics about the survey, survey cut off was established at eight weeks after initial email.

Figure 1:

- Note: this conceptual model does not align. The overall behaviour is PA promotion but you are describing the attitude and subjective norm towards PA in general. These are two different behaviours. In the text you describe subjective norms related to PA promotion, but not in the figure. The attitude should be “attitudes in respect to PA promotion for the cancer population”…do they think it is important / helpful, etc. Must all link together and make sense.

• This has been re-worded and we hope it addresses your comments.

Good description of initial validation.

• Thank you

Line 212-215 should go in ‘recruitment’ section. 

• This has been relocated.

It is confusing how it is laid out…I think you are describing two separate studies. The methodology section should only be in reference to this study.

• This has been

Paragraph Lines 219-231: you are presenting results here. Should go in results section. 

• Moved to the results section

Same thing for paragraph starting Line 233.

• Moved to results section

Organize methods / results more clearly to delineate the steps that were taken. It is all important, just needs to be organized more appropriately. 

• This section has been arranged so that the validation of the study instrument is at the start of both the methods and the results section. This should ease the flow of the paper for the reader.

Results:

Line 248: how much needed to be complete to be included? All questions? Confirm in methods.

• This has been added to the results: Only fully completed surveys were included, fifteen responses were excluded due to incomplete/ missing data, leaving a total of 113 completed for analysis. From total responses, 111 met inclusion criteria (having consulted a cancer patient in the last 12 months).

• This has been added to the methods: The analyses were conducted on SPSS V25 (43) on completed surveys.

Line 255: add ‘exercise’ before intensity. Also, I think you are missing a statement about % who take part in exercise prior to this sentence.

• Added 

Line 280-281; this should go in the methods. Use sub-headings in the results section to make clear to readers.

• This has been moved

• Headings have been added to the results section

Line 293-295; can you describe this further…for example, what Gender? How many years practicing? This information will be interesting for readers but isn’t stated anywhere.

• This information can be found in table 1 – this has now been added so that the reader can find it easier.

Line 311-312; what there a cut off value for this? How much did practitioners have to exercise to be more likely to predict…any?

• Over 150 minutes of exercise was used as a cut off for moderately active participants, this has been changed in the manuscript to reflect this

Discussion:

Line 362-363: you should link this finding to limitations also; perhaps those who responded to your recruitment email were particularly interested in this topic because they were already regular exercises; those who didn’t think it was important didn’t respond. Add into limitations also.

• This has been added to both the discussion and limitations as requested

Paragraph starting line 369. I think it was also interesting that only 19% were aware of the new COSA guidelines. This was a major publication in exercise oncology globally! So, the fact that GP’s within Australia are not aware of it is surprising.

• Yes we also found this surprising however, we so also see this clinically.

Line 418 and throughout discussion: do not need to reference tables in discussion sections

Throughout discussion section you may want to refer to other studies looking at barriers/facilitators to exercise promotion for this population. What gets in the way and how does this relate to the TPB components? How can this be addressed?

• Reference has been made to a number of other studies as suggested see line 465, 471, 495, 499, 500, 504, 508, 514.

• Table 5 has been removed and references to tables have been removed – further details and references have been added to discuss findings from other studies.

Summary:

Overall, an interesting topic. Thank you for taking the time to conduct research in this area. Needs some edits for clarity before publication.

I am concerned around how the TPB was described and how the conceptual model include two different behaviours (PA and PA promotion)...this needs to be clarified.

More detailed and clear information needed in methods; results were presented in methods as well.

Increase clarify of results (give more specific information).

• Thank you for your amazing feedback, it has improved the manuscript considerably – the authors hope that with the suggested additions that you will be happy with the changes.

Reviewer #2: Thanks for the opportunity to review this paper, it was interesting to read. Please see comments which are a mix of relevant points and pedantry.

Abstract

I wonder if the term 'cancer patients' is appropriate these are people with a diagnosis of cancer and that label really does jar when I read it. We probably don't label patients as 'diabetics' or schizophrenics' any linger so maybe the term 'patients with cancer' or perhaps 'patients living with cancer', although not exactly mellifluous. would be more appropriate? Same comment applies to 'the cancer population' and other such terms

• Thank you for your very thoughtful comment, this has been changed as suggested

Background

Line 85 33% of cancer related deaths are attributable to physical inactivity....' I wonder is this statement too strong. I agree that physical inactivity and poor diet are contributory factors but as an absolute causal link it's very difficult to be that categorical. Could you say that they are significant contributory factors?

• This has been changed and now reads: Physical inactivity and poor dietary behaviours are significant contributing factors to cancer-related deaths

Line 93 -'It has been established that patients who have been told to be active by their HCPs’ have improved levels and adherence levels' this doesn't make sense to me improved levels and adherence level ? Im presuming you mean persistence with the exercise programme? Maybe reword ?

• This has been changed and now reads: It has been established that patients who have been told to be active by their HCPs’ have improved exercise and adherence levels

Line 103 - 'in this case what other HCP’s physical activity believe and promotion of PA to their cancer patients...' consider reword doesn't make sense should that beliefs are?

• This has been changed to: The second construct is subjective norms which is defined by the perceived belief of what others think about a certain behaviour, in this case what other HCP’s believe about the promotion of PA to their patients living with cancer

Line 105 repetition of 'perceived' not sure second one is necessary and removing would improve sentence flow

• Thank you, that has been changed to: The final construct, titled perceived behavioural control, questions beliefs on the difficulty to perform the behaviour, in this case providing PA recommendations

Line 106 should read 'individuals''?

• Thank you, that has been corrected

Line 109 - define acronym if you are going to use it in the rest of the document Clinical Oncology Society of Australia (COSA)....

• This has now been added

Line 112 should read 'the general population'

• This has been added

Line 118 '...treatment, which cancer...'

• Added 

Line 122 - could these two sentence be combined, second sentence is partially repeating the first.

• The first sentence was deleted with more information added to the second sentence, it now reads: During treatment in Australia, it is reported that patients contact time is spent predominantly with oncology nurses and less time with their oncologists

Line 126 is a possessive so should read 'HCPs'"

• This sentence and the sentence below has been changed and now reads: HCPs attitudes towards PA during the cancer treatment phase is evident (16, 24, 36), however, there is limited understanding of this during the post treatment phase (23, 37-39), GPs views and role within this area has not been explored.

Line 133 should read '...Australian GPs. which ...'

• A comma has been added as I think that is what you meant

Line 138 you have used GPs elsewhere, why write out in full here?

• This has been changed

Methods

I'm not convinced that you need to specify whether data is ordinal, nominal or scalar. Its repetitive and doesn't help the flow of the text. I would argue anyone reading this is capable of that distinction and the nature of the data in statistical terms is immaterial, it is what it is.

• Thank you, this has been removed.

Results

Line 258 - surprising that you only had people identify as male or female and there were no other options. Did you provide options for gender or was your question about sex ie male/female as options? If so this should read sex.

• Thank you, this has been changed

Line 296 should read recommended 'PA' ....

• Changed 

Line 311 - personal preference but as a reader where percentages are quoted I often like to see the number, just for context as I read so 69% (n=22) or 22 GPs (x%), whichever works best for you.

• This has been added

Discussion I would argue that Table 5 should be in the results section and discussed in the context of other literature in the discussion. If you feel uncomfortable with the results of the literature review from other HCPs then remove this and discussion in the discussion section. Feels unusual to see a table presented like this in the discussion.

• This has been removed from the manuscript

Line 396 should read 'Conversely, on average....'

• This has been added.

Line 390 should read ...in PA, which....'

• This has been added

Line 404 should read 'In the current study, only 32%....'

• This has been added

Line 430 is adverse effects a better term?

• This has been changed as suggested

We would normally expect to see a conclusion in a study of this type, I think it would add clarity to the final part of this paper, which does wander a little bit.

• This has been added

Thank you for your very thoughtful comments, they have made a big difference to the paper.

Reviewer #3: I will focus on methods and reporting. Statistical analyses are appropriate.

Major

1) State clearly all information in the data analysis section, all used variables. How was correct and incorrect PA knowledge determined? Clarify what the outcome is and how it is recorded.

• This has been changed throughout the manuscript, we hope that they changes improve the clarity and flow of the paper.

Minor

1) Abstract: more clarity on regression modelling, no information on covariates of interest

• This has been changed and added to the abstract

2) the survey is potential problematic, as the authors acknowledge. the generalisability of the survey is questionable.

• Thank you – yes we have added this to our limitations

3) in table 4 better to report CIs rather than SEs

• The authors believe that due to the categorical nature of the analysis it would be difficult to report the range of CI values. SE was used to measure the accuracy of the estimates equal to the SD of the distribution which is more beneficial than proposing a range with the large number of values for each parameter. An example is provided below for the gender variable in % recommended PA:

• 1 = male, 2 = female

 What gender do you identify as? Statistic Std. Error

What percentage of your cancer patients have you recommended physical activity to? 1 Mean 3.60 .199

 95% Confidence Interval for Mean Lower Bound 3.20 

 Upper Bound 4.00 

 5% Trimmed Mean 3.61 

 Median 4.00 

 Variance 1.855 

 Std. Deviation 1.362 

 Minimum 1 

 Maximum 6 

 Range 5 

 Interquartile Range 2 

 Skewness -.238 .347

 Kurtosis -.604 .681

 2 Mean 4.38 .183

 95% Confidence Interval for Mean Lower Bound 4.01 

 Upper Bound 4.74 

 5% Trimmed Mean 4.43 

 Median 5.00 

 Variance 2.143 

 Std. Deviation 1.464 

 Minimum 1 

 Maximum 6 

 Range 5 

 Interquartile Range 3 

 Skewness -.529 .299

 Kurtosis -.956 .590

4) Consider other graphical outputs to present your results and make them more accessible

• Thank you, we have added sub-headings to assist with the flow of the results section, it is now much easier to follow

6. PLOS authors have the option to publish the peer review history of their article (what does this mean?). If published, this will include your full peer review and any attached files.

Do you want your identity to be public for this peer review? For information about this choice, including consent withdrawal, please see our Privacy Policy.

Reviewer #1: No

Reviewer #2: No

Reviewer #3: No

---

## [Decision Letter · Decision Letter 1]

20 Oct 2020

Physical Activity for People Living with Cancer: Knowledge, Attitudes, and Practices of General Practitioners in Australia

PONE-D-20-10478R1

Dear Dr. Toohey,

We’re pleased to inform you that your manuscript has been judged scientifically suitable for publication and will be formally accepted for publication once it meets all outstanding technical requirements.

Kind regards,

Adam Todd, PhD

Academic Editor

PLOS ONE

Additional Editor Comments (optional):

Reviewers' comments:

Reviewer's Responses to Questions

**Comments to the Author**

1. If the authors have adequately addressed your comments raised in a previous round of review and you feel that this manuscript is now acceptable for publication, you may indicate that here to bypass the “Comments to the Author” section, enter your conflict of interest statement in the “Confidential to Editor” section, and submit your "Accept" recommendation.

Reviewer #3: All comments have been addressed

2. Is the manuscript technically sound, and do the data support the conclusions?

Reviewer #3: Yes

3. Has the statistical analysis been performed appropriately and rigorously? 

Reviewer #3: Yes

4. Have the authors made all data underlying the findings in their manuscript fully available?

Reviewer #3: Yes

5. Is the manuscript presented in an intelligible fashion and written in standard English?

Reviewer #3: Yes

6. Review Comments to the Author

Reviewer #3: The authors have responded to the points raised satisfactorily. My last point in the previous iteration related to thinking about graphs to present the findings. Currently the paper is lacking in that area. The authors misunderstood that point last time.

7. PLOS authors have the option to publish the peer review history of their article (what does this mean?). If published, this will include your full peer review and any attached files.

Reviewer #3: No

---

## [Editor Report · Acceptance letter]

26 Oct 2020

PONE-D-20-10478R1 

Physical Activity for People Living with Cancer: Knowledge, Attitudes, and Practices of General Practitioners in Australia 

Dear Dr. Toohey:

I'm pleased to inform you that your manuscript has been deemed suitable for publication in PLOS ONE. Congratulations! Your manuscript is now with our production department. 

Kind regards, 

on behalf of

Dr. Adam Todd 

Academic Editor

PLOS ONE